# A sustained change in the supply of parental care causes adaptive evolution of offspring morphology

Benjamin J.M. Jarrett[1,5], Emma Evans[1], Hannah B. Haynes[1], Miranda R. Leaf[1], Darren Rebar [1,2], Ana Duarte[1,3], Matthew Schrader[1,4] & Rebecca M. Kilner[1]

Although cooperative social interactions within species are considered an important driver of evolutionary change, few studies have experimentally demonstrated that they cause adaptive evolution. Here we address this problem by studying the burying beetle *Nicrophorus vespilloides*. In this species, parents and larvae work together to obtain nourishment for larvae from the carrion breeding resource: parents feed larvae and larvae also self-feed. We established experimentally evolving populations in which we varied the assistance that parents provided for their offspring and investigated how offspring evolved in response. We show that in populations where parents predictably supplied more care, larval mandibles evolved to be smaller in relation to larval mass, and larvae were correspondingly less self-sufficient. Previous work has shown that antagonistic social interactions can generate escalating evolutionary arms races. Our study shows that cooperative interactions can yield the opposite evolutionary outcome: when one party invests more, the other evolves to invest less.

[1] Department of Zoology, University of Cambridge, Downing Street, Cambridge CB2 3EJ, England. [2] Department of Biological Sciences, Emporia State University, Emporia, KS 66801, USA. [3] College of Life and Environmental Sciences, University of Exeter, Penryn Campus, Cornwall TR10 9FE, England. [4] Department of Biology, University of the South, Sewanee, TN 37383, USA. [5]Present address: Department of Entomology, Michigan State University, East Lansing, MI 48824, USA. Correspondence and requests for materials should be addressed to B.J.M.J. (email: bjmjarrett@gmail.com)

Social behaviour drives evolutionary change by exposing traits in one individual directly to selection by a social partner[1–3]. For example, antagonistic interactions during reproduction have long been recognised as a powerful force of selection on animal morphology[1,4,5], accounting for the evolution of male weaponry[6,7] and ornamentation[8], and females that are larger than males[9]. Within animal families, an evolutionary conflict of interest between parents and offspring can explain why dependent young bear exaggerated structures and perform intense, multimodal offspring begging displays[10]. The majority of previous work has thus emphasised how social interactions in animals can drive evolutionary change through arms races, in which a rival's more effective weaponry[11] or a receiver's greater sales resistance[12] causes traits to become increasingly exaggerated.

Yet social interactions between animals also comprise acts of cooperation, especially during reproduction. Here social partners work together for some shared fitness benefit[13]. The more one individual contributes to this shared endeavour, the lower the burden on its cooperative partner to make a contribution. Thus whereas antagonistic interactions can drive the evolution of ever greater trait investment in each party, when a partner invests more in a cooperative interaction it can select for reduced trait investment in its social partner. Nevertheless, relatively little work has tested this idea experimentally. Previous studies have described negative genetic and phenotypic correlations between traits in social partners that are consistent with selection resulting from acts of cooperation (e.g., refs. [14–16]). However, direct evidence that greater acts of cooperation cause an evolved reduction in traits in the social partner has seldom been obtained before.

We addressed this problem with experiments on the burying beetle *Nicrophorus vespilloides*. Burying beetles commonly exhibit elaborate biparental care, centred on the carcass of a small vertebrate. Parents convert the carcass into an edible nest for their larvae by removing the fur or feathers, covering the flesh in antimicrobial exudates, rolling it into a ball and burying it in a shallow grave[17–19]. The larvae hatch from eggs laid nearby in the soil and crawl to the carcass. Parents assist the newly-hatched larvae in penetrating the carcass, and colonizing it, by biting small incisions in the flesh. Once the larvae have taken up residence upon the carcass, parents may stay to defend them and to feed them via oral trophallaxis[20]. Larvae also feed themselves from the carrion and can survive without any post-hatching care[20,21]. The experiments we report here focused on these acts of parent-offspring cooperation and their selective influence on larval morphology. We began by documenting variation in wild-caught beetles, both in their supply of care when bred in the lab and in the relative size of the larval mandibles. We used descendants of beetles caught from the same wild populations to found experimental populations (see 'Methods'), upon which we imposed two markedly different levels of parental care. By tracking these populations across the generations, we determined how larvae, and their parents, evolved and adapted in response to manipulated variation in the extent of parent-offspring cooperation. We find that when parents were present to help their offspring feed on the carcass, the larvae evolved relatively smaller mandibles. Conversely, when parents provided no care and larvae had to self-feed, their mandibles were relatively larger.

## Results and discussion

**Variation in wild-caught beetles: parental care**. In our previous work, we observed that the provision of post-hatching care is highly variable, with either parent leaving the brood at any time, from soon after larval hatching to larval dispersal from the carcass[22]. Similar observations have been made by other researchers,

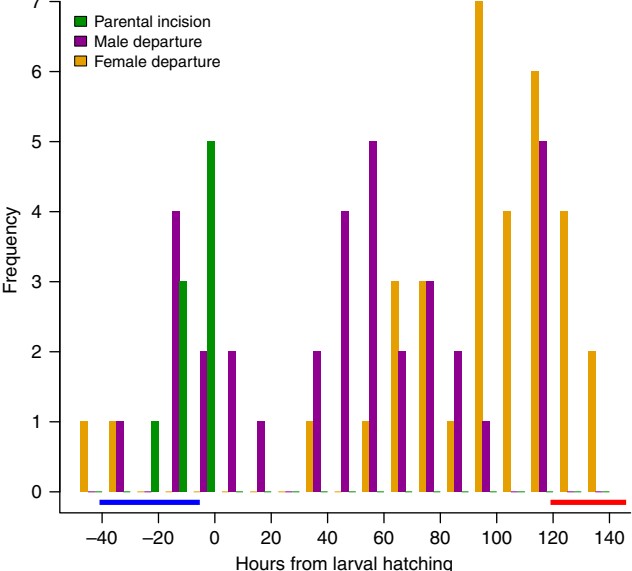

**Fig. 1** Variation in the duration of parental care in wild-caught individuals. Variation in the duration of maternal (orange bars) and paternal (purple bars) care, and the timing of biting the feeding incision in the carcass by wild-caught parents under laboratory conditions (green bars, data shown only for 9 pairs that inserted an incision prior to larval hatching). Data are scaled relative to the timing of larval hatching at 0 h. n = 34 pairs. Horizontal bars indicate when we removed parents in the No Care (blue bar) and Control (red bar) treatments. Note that each treatment reduces variation in the extent of parental assistance supplied to larvae, both in biting the feeding incision and caring for offspring after hatching

both in the laboratory[23] and in field studies of *N. vespilloides*[24] and in other burying beetle species[25]. We quantified variation in the supply of care, under standardized conditions, by breeding wild-caught individuals in the laboratory (see 'Methods'). We bred 34 pairs, each in a box with a one-way exit port, through which the adults were free to leave at any time but could not return[22], and noted the time of parental departure. While parents are caring for offspring they spend virtually all their time on, or very close to, the carcass. Previous studies have shown that once individuals stop caring in nature, they move away from the carcass and do not return[25]. Furthermore, even when parents were given the opportunity to return to their brood in laboratory experiments, they did not provide substantial levels of care[26]. Therefore it is extremely unlikely that actively caring parents accidentally wandered through the exit port in our experiments.

We found considerable continuous variation in the duration of parental attendance at the carcass, in both males and females (Fig. 1). At one extreme, both parents left before the larvae hatched in two of the 34 breeding attempts, whereas in five cases both parents stayed until larvae dispersed away to pupate. We also observed qualitative differences in carcass preparation. In nine cases, we noted that parents had made a feeding incision in the carcass before their larvae hatched (Fig. 1), but in the remaining 25 breeding attempts parents delayed biting a feeding incision until after their larvae had hatched. Parents potentially assist larvae in gaining access to resources on the carcass in two key ways: they bite a feeding incision, presumably so that newly-hatched larvae can more easily penetrate the carcass and feed upon it; and they provision offspring directly. Our experiment found variation in both forms of parental care in wild-caught beetles, bred under standard conditions in the laboratory.

**Variation in wild-caught beetles: larval mandibles**. The high level of variation in the duration of parental attendance at the carcass, and in the extent to which the carcass is prepared prior to larval hatching, means that larvae may receive no parental assistance in gaining access to the resources on the carcass. The larval trait most likely to influence their performance, independently of the parents, is the larval mandibles because they are essential for gaining access to the carrion nest and consuming it. We began by testing whether larvae cope with variable levels of parental care by exhibiting adaptive phenotypic plasticity in the relative size of their mandibles. We paired wild-caught individuals and bred them in the laboratory under standard conditions (see 'Methods'). We exposed larvae to one of two different social environments, lying close to each extreme of the continuous variation in parental assistance that we documented in our first experiment (illustrated on Fig. 1). At one extreme, we created a 'No Care' environment by removing parents after carcass preparation was complete but before the larvae had hatched. At the other extreme we made a 'Control' environment by keeping parents within the breeding box so that they remained with their young until larval dispersal, and were able to provide care during this time. Although the level of care provided by parents in the Control environment was highly variable, parents still provided substantially more care for their young on average than those in the No Care populations (see below). We predicted that if mandibles exhibited adaptive phenotypic plasticity then they should be relatively larger for a given body size in the No Care environment than in the Control environment, to compensate for the lack of post-hatching parental assistance.

We collected third instar larvae from each treatment and weighed them. Then we dissected out the mandibles, mounted them, and measured their length (see 'Methods and materials', Supplementary Fig. 1). Although selection on the mandible is likely acting on first-instar larvae, we focused on the third instar to maximise precision in measuring both mandible length and body mass. This approach is justified by two observations. First, *Nicrophorus* beetles show consistency in morphology across instars: a larva with relatively large mandibles in its first instar will also have relatively large mandibles in its third instar[27,28]. This means that morphological adaptations that facilitate self-feeding in first instar larvae will be apparent across larval development. Second, developmental and genetic studies of other insects indicate that the mechanisms governing mouthpart size do not differ between larval instars[29]. Thus, we should be able to observe the correlated evolution of the mandibles of the third instar when selection has acted on the developmental mechanisms that dictate mandible size in the first instar.

We found no evidence for adaptive phenotypic plasticity in the mandible size of offspring from wild-caught parents. The scaling relationship between mandible size and body size did not differ significantly between the two different care treatments (ordinary least squares regression, OLS: $t_{103} = -1.21$, $P = 0.23$; major axis regression, MA: LR = 1.55, $P = 0.21$, Fig. 2). Furthermore, whether raised in a No Care or the Control treatment, the slope of scaling relationship between mandible size and body size (i.e., the allometric slope, $\beta$) did not differ significantly from 0 (OLS regression, Control: $t_{52} = 1.60$, $P = 0.12$; No Care: $t_{52} = -0.11$, $P = 0.91$; Combined populations: $t_{106} = 1.58$, $P = 0.12$. MA regression, Control: $r_{52} = 0.22$, $P = 0.12$; No Care: $r_{52} = -0.02$, $P = 0.91$; Combined populations: $r_{106} = 0.15$, $P = 0.12$, Supplementary Table 1). We found instead a high level of variation in the relationship between mandible size and larval mass. Furthermore, on average, larval mandibles were consistently large, irrespective of larval mass (Fig. 2). One interpretation of this result is that by maintaining relatively large mandibles for self-feeding, larvae are adapted on average to anticipate the worst possible scenario of

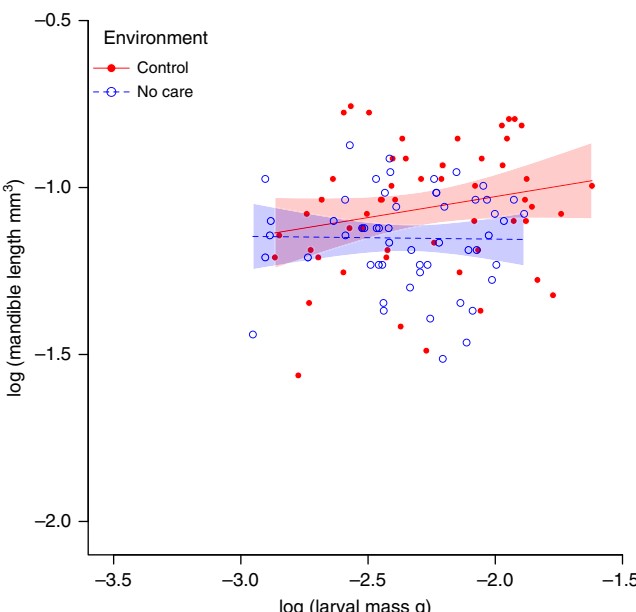

**Fig. 2** Larval mandible allometry in offspring of wild-caught parents. The allometric relationship between larval mandible length and larval body mass in the offspring of wild-caught parents. Larvae were either raised either in a Control environment (red filled datapoints, red solid line, $n = 54$) or a No Care environment (blue open datapoints, blue dashed line, $n = 54$). Ordinary least squares regression lines are shown with 95% confidence intervals

receiving no parental assistance at all in accessing the resources on the carcass.

**Experimental evolution**. We exploited the high level of natural variation in parental care, and in the relative size of the larval mandibles, to establish populations of burying beetles in the laboratory ($n = 4$) that we subjected to experimental evolution. We imposed two different but predictable regimes of parental care, applied experimentally at each generation. Two populations experienced the Control treatment, which was identical to the Control treatment in the previous experiment on wild-caught individuals. Here both parents were left with their offspring throughout larval development. Most broods in this treatment received at least 24 h of maternal care (Fig. 1). Parents also cut a feeding incision cut into the carcass for their brood (Fig. 1). The remaining two populations experienced the same No Care treatment as described above. All broods in this treatment predictably experienced no post-hatching care at all (see 'Methods').

**Experimentally evolved adaptations in parents**. To determine the likelihood that parents in this treatment cut a feeding incision in the carcass for their brood, we assayed the populations after 13 generations of experimental evolution. We found that No Care parents had evolved to be more likely to insert a feeding incision in the carcass before we removed them experimentally (see Supplementary Materials, Supplementary Fig. 2). Specifically, parents from the No Care populations were approximately twice as likely to make an incision into the carcass prior to larval hatching, than individuals from either the Control populations or the Wild populations (binomial GLM: $z = 5.28$, $P < 0.001$). Therefore, after 13 generations of experimental evolution, parents predictably cut a feeding incision in the carcass for their broods in the No Care populations.

Next, we investigated whether this feeding incision functioned to promoted larval fitness (see 'Methods'). We allowed pairs of beetles from a stock laboratory population to prepare a carcass and removed the adults before they could make an incision. We cut a small feeding incision in half the carcasses ourselves, keeping control carcasses without an incision. Then we added ten newly-hatched larvae to each type of carcass and let them develop without any parental care. We found that larvae survived better on a carcass when we had cut a small feeding incision in it, than when we had not (binomial GLM: $z = 9.07$, $P < 0.001$, Supplementary Fig. 3), thus replicating the results of a previously published experiment[30]. Therefore, by advancing the time at which they bit a feeding incision in the carcass, No Care parents promoted their offspring's fitness and in this way adapted to the No Care social environment that we had imposed experimentally. Even though larvae in the No Care populations received no post-hatching care, they did experience some degree of parental assistance in gaining access to resources on the carcass, through the feeding incision bitten by their parents.

**Experimentally evolved adaptations in offspring**. The level of parental assistance received by larvae in each experimental population was thus more predictable than observed in wild burying beetles, even though the mean level of care supplied differed between the experimental treatments (see Fig. 1, 'Methods'). We expected that consistent exposure to a more predictable environment after hatching should induce an evolutionary change in the relative size of the larval mandibles, according to the level of care supplied. Specifically, selection on larvae to be self-reliant, and maintain relatively large mandibles, should be relaxed in our experimental populations because here larvae could depend on parents for at least some degree of assistance in accessing the resources on the carcass. Our expectation was that in the experimental populations we should see a corresponding evolved change in the scaling relationship between the larval mandibles and larval size: the greater the supply of predictable parental assistance, the more strongly larval mandible size should scale with larval mass.

To test this prediction, we measured the gradient ($\beta$) of the scaling relationship between mandible size and larval size in the two experimental populations. Before measuring mandible size, we put each population through a common garden environment to eliminate any potentially confounding environmental effects (see 'Methods'). We found that $\beta$ was now significantly positive for both experimental populations, unlike the ancestral wild population, such that larval mandible length now increased with larval mass. Furthermore, $\beta$ differed between experimental populations (Supplementary Table 2, Fig. 3), and in proportion to the level of parental assistance (interaction between experimental care regime and larval mass on larval mandible length, OLS: $t_{163} = -2.87$, $P = 0.005$; MA: Likelihood ratio $= 9.65$, $P = 0.002$, Fig. 3). Larvae from the Control populations could rely on extensive parental assistance in penetrating the carcass, and they had relatively smaller mandibles. Larvae from the No Care populations could depend on less parental help, and they had relatively larger mandibles. Importantly, neither population exhibited the scaling relationship that we found in wild-type larvae, which had even larger mandibles on average for a given body size (Fig. 2). In general, exposure to different levels of predictable parental care drove the evolution of new larval mandible scaling relationships in both the Control populations and the No Care populations. Because some level of parental care was supplied predictably in all the experimentally evolving populations, there was no risk that the larvae in these treatments would ever experience the worst-case scenario of no feeding

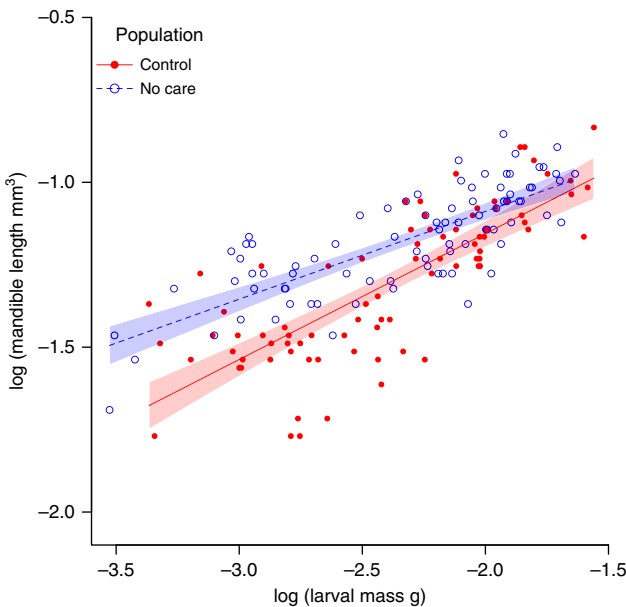

**Fig. 3** Larval mandible allometry in the experimentally evolving populations. The allometric relationship between larval mandible length and larval body mass in the offspring from experimental populations evolving in a Control environment (red filled datapoints, red solid line, $n = 82$) and a No Care environment (blue open datapoints, blue dashed line, $n = 86$). Ordinary least squares regression lines are shown with 95% confidence intervals

incision in the carcass and no post-hatching care. By contrast, in wild populations, where the supply of care is far more unpredictable, this is an outcome that some larvae can experience (Fig. 1). Perhaps larvae in wild populations maintain such relatively large mandibles as a conservative bet-hedging strategy, and this explains why they are larger than those seen in the experimentally evolving populations.

Finally, we investigated whether the new scaling relationships in the Control and No Care populations were adaptive. Specifically, we asked whether smaller larvae from the No Care populations were more likely to survive in a No Care environment than smaller larvae from the Control environment. We focused particularly on smaller larvae because the evolved difference between populations in relative mandible size was most pronounced in this subset of individuals (Fig. 3).

We added broods of ten larvae, drawn either from the Control population or the No Care population, onto a carcass prepared by stock beetles. We measured how well smaller individuals survived when given no assistance in penetrating the carcass, and no post-hatching care. We predicted that smaller larvae from the No Care populations would have a greater chance of survival under these social conditions than smaller larvae from the Control populations because the No Care larval mandibles were relatively larger. Overall, we found that more offspring survived from the No Care populations than the Control populations (binomial GLM: $z = 4.02$, $P < 0.001$). This shows that larvae from the No Care populations were better adapted to a No Care environment than larvae from the Control populations. Furthermore, the smallest survivor from each brood was indeed smaller in the No Care populations than in the Control populations (GLM: $\chi^2 = 4.16$, $P = 0.04$, Fig. 4). This result is consistent with the possibility that smaller larvae are more likely to survive without post-hatching care when their mandibles are disproportionately large. Nevertheless, further work is required to determine how much of the increase in larval survival is due to the mandibles alone. The scaling relationship between the larval mandibles and larval size is not the only larval adaptation to have evolved in the No Care

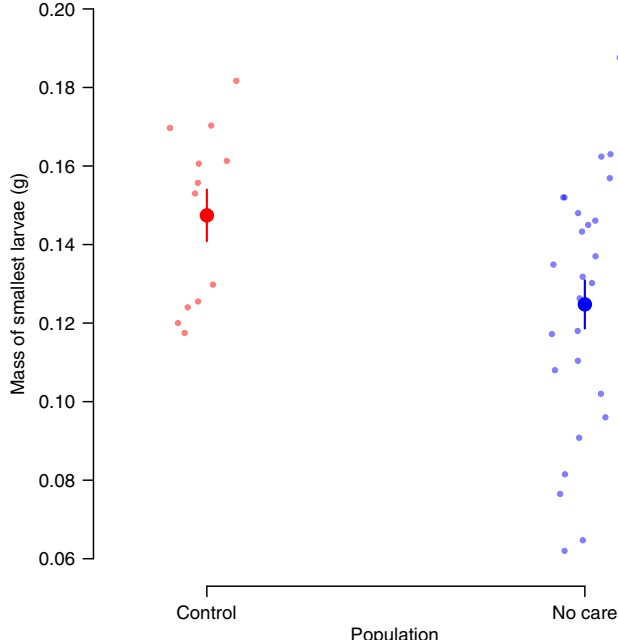

**Fig. 4** The effect of experimental evolution in the Control and No Care environments on the size of the smallest surviving larva. The mass of the smallest surviving larva in the brood, in relation to the social environment experienced by the experimentally evolving populations, when broods of 10 larvae were left on a carcass prepared by stock beetles with no incision and no post-hatching care (Control: $n = 12$; No Care: $n = 27$). Means with standard errors are shown

populations. In separate work, we have found that other larval adaptations also contribute to larval survival during development in a No Care environment, including more synchronised egg hatching within clutches[31].

To sum up, we found that in wild populations of burying beetles larval mandible size could not be predicted by larval size, and the supply of parental care was also highly variable. We suggest that the high level of variation in larval morphology can be explained by the highly variable levels of parental assistance on offer to larvae (Fig. 1, refs. [23,26]). When we enforced more predictable levels of care experimentally, we evolved larvae with mandibles that scaled much more predictably with their size (Fig. 3). However, the mechanism that maintains high levels of variation in this scaling relationship within wild populations (Fig. 2) remains to be determined in future work. We cannot tell from our data whether it is due to variation within broods, or among them. Variation within broods could exist as a bet-hedging adaptation for an unpredictable supply of care. Alternatively, variation among broods could arise if parents induce appropriate offspring adaptations for the extent of care they intend to provide (see ref. [32] for an example from birds). Whatever the mechanism at work, the high level of variation in relative mandible size sequestered in natural populations explains in part why we were able to detect evolutionary change so rapidly in our experimental populations.

More generally, our experiments show how cooperative interactions can influence morphological evolution in a social partner. In our experimental populations, we varied the degree to which parents helped their larvae so that they consistently provided either relatively little assistance (only biting a feeding incision in the No Care populations) or substantially more (biting a feeding incision and provisioning for at least 24 h in the Control populations). This, in turn, changed the optimal size of the larval mandibles. The greater the extent of parental assistance, the greater the cost to smaller larvae of maintaining relatively large, relatively redundant mandibles—and the greater the strength of selection to produce smaller mandibles. By predictably contributing more to larval nourishment, across the generations, parents caused an evolved reduction in larval traits for self-feeding. Previous work has shown that antagonistic social interactions can generate escalating evolutionary arms races: when one party evolves to invest more in a social trait it provokes the antagonist to invest an even greater amount (e.g., refs. [7,11,33,34]). Our study shows that cooperative interactions can yield the opposite evolutionary outcome. When one party evolves to invest more, the cooperating partner evolves to invest less.

## Methods

**Predictability of parental care by wild-caught individuals.** Previous work suggests that in nature, the duration of parental attendance at the carcass by burying beetles is highly variable. For example, when carcasses in the wild were exhumed, *N. orbicollis* parents were both absent in almost 10% of cases[35]. We measured variation in the duration of care supplied by wild-caught individuals, in a controlled laboratory environment. We caught beetles from two natural populations (Gamlingay Woods and Waresley Woods) in Cambridgeshire, UK and kept individuals under identical conditions for one week in order to standardise their condition before breeding. Individuals were randomly assigned a breeding partner within their respective population of origin. Those caught in the same trap were not bred together to prevent possible inbreeding. For breeding, a pair was placed in a large breeding box (28.5 × 13.5 × 12 cm) that had been divided into two sections, one twice as the large as the other, as described in ref. [22]. The partition had a hole cut into it, with a tube and cloth tunnel on one side. The larger section housed the adults and carcass and was lined with commercially bought compost. Parents could leave this breeding chamber at any point and enter the smaller section but the cloth tunnel acted as a one-way valve and prevented re-entry[22]. In this way, we allowed parents to tend their offspring for a period of their choosing, just as they would in nature. We could also time the duration of their attendance at the carcass. Pairs were given a recently defrosted mouse carcass (8–12 g) to initiate breeding. The boxes were left in the dark and checked four times a day to measure the time of departure for each parent. Departed beetles were removed from the smaller section of the box. Towards the end of carcass preparation, we also checked the carcass carefully for the presence of an incision cut by parents, to facilitate penetration of the carcass by newly-hatched larvae. The timing of larval hatching was noted. The time when the feeding incision was cut and when parents left the carcass were each scaled relative to this event. Departure times for males and females within each pair were compared with a *t*-test. As found previously in other burying beetle species[25], males left earlier than females ($t_{66} = 4.07$, $P < 0.001$).

**Larval mandibles and body size in wild beetles.** We caught beetles from two natural populations (Gamlingay Woods and Waresley Woods) in Cambridgeshire, UK. These two populations contributed descendants to the population used to found the experimental populations described below. They were bred in a standard breeding box (17 × 12 × 6 cm). Larvae experienced two different social environments, Control and No Care. In the Control treatment, parents were left to care and interact with their larvae throughout development. In the No Care treatment, parents were removed at ~53 h after pairing, before the larvae hatched, so that there were no interactions at all between parents and offspring[26,36]. Eight days after pairing, when the third instar larvae were dispersing away from the carcass, they were collected from both treatments and stored at −20 °C. Two larvae per brood were randomly chosen for measurement, to ensure we had replicate measures from each brood. Dead larvae were subsequently weighed while still frozen. Wet larval mass and dry larval mass are highly correlated ($n = 53$, $r = 0.96$, $P < 0.001$). We used wet larval mass in all analyses. Larval mandibles were dissected from third-instar larvae under a dissection microscope using two entomological pins. One mandible was then isolated and mounted in nail polish to ensure it laid flat for measurement, which was done blind to the treatment and to the mass of the larva. Mandible length and width were measured using a Weiss graticule eyepiece, after calibration (Supplementary Fig. 1). They were highly correlated ($n = 106$, $r = 0.63$, $P < 0.001$). Our analyses focus on mandible length because previous work suggests it is tightly linked to mandible function. For example, ants with longer mandibles have been shown to create larger incisions during foraging[37].

We compared the scaling relationship (static allometry) between larval mandible length and body mass for third instar larvae raised in the Control and No Care environments. Static allometry is defined as $y = \alpha x^{\beta}$, where $y$ is the size of the trait of interest, $x$ is body size, $\alpha$ is the allometric intercept, and $\beta$ is the allometric scaling parameter. Taking the natural logarithm of the trait size and body size yields a linear relationship, $\log(y) = \log(\alpha) + \beta \log(x)$, where $\log(\alpha)$ is the intercept and $\beta$ is the slope of the line. Morphological traits scale isometrically (geometrically similar) when $\beta = 1$; that is, they scale in proportion to each other[38–40]. A negative allometry arises when $\beta < 1$. This arises, for example, when smaller individuals bear a trait that is disproportionately large for their body size.

The statistical techniques used for quantifying static allometry have been the subject of some debate. Some authors have argued that since both the predictor variable (body size) and response variable (mandible length) are measured with some error, MA is the best approach for estimating allometric slopes[41,42]. Another statistical technique used to analyse allometry related to MA regression is standardised (or reduced) major axis regression (SMA). This involves calculating the ratio of standard deviations between both variables. However, it does not perform well when the scaling slope is close to or equal to zero, and therefore we did not use it here. Other authors have argued that OLS is a better approach[43–45]. We used both MA and OLS approaches to analyse the data. We analysed data from each treatment separately, and also analysed the whole dataset ('Combined' data).

For all allometric analyses, all variables were ln-transformed. Mandible length was cubed prior to analysis so that measures on both axes were in cubic units. This made it easier to interpret the slope estimates because, with length and mass both in cubic units, an isometric relationship can be found when $\beta = 1$. We used R 3.3.0[46]: the MA analysis was performed using the package smatr[46]. Both OLS and MA estimates are given with the 95% confidence intervals and each output is displayed in Supplementary Tables 1 and 2. Figure 2 shows the results from the more conservative OLS analysis. Using MA analysis did not qualitatively change any of the results. Note that, since we analysed only two larvae per brood, we cannot tell from our data whether larval mandibles scale differently with body size within families, or whether different families have mandibles that exhibit different scaling relationships with body size.

**Experimental evolution**. The experimental populations were established from a pool of 671 individuals descended from 32 pairs of wild beetles collected from four localities in Cambridgeshire, UK (Byron's Pool, Gamlingay, Waresley, and Overall Grove), within approximately 20 km of each other (straight line distance)[47]. Of the original 64 wild beetles, 14 were from Gamlingay Woods, while 5 from Waresley Woods. From the pool of 671 individuals, the two Control populations were founded with 35 pairs of haphazardly chosen individuals, and the No Care populations were founded with 50 pairs of haphazardly chosen individuals. There is more variation in relative mandible size within populations than among populations. In other words, larvae are highly variable in their relative mandible size, from whichever Cambridgeshire population we collect them. Therefore we think it is reasonable to assume that the measurements we made from Gamlingay and Waresley Woods larvae (above) are representative of the ancestral population. We further assume that the relative size of larval mandibles at the start of experimental evolution was similar in all four experimental populations, though there is a very low theoretical possibility that this was not the case.

At each generation, larvae were consistently exposed to either Control or to No Care, exactly as described above. Each type of social environment was replicated twice, generating four experimental populations in all: two were Control (F1 and F2) and two were No Care (N1 and N2). The first replicates (F1 and N1) were scheduled to breed a week before the second replicates (F2 and N2), to spread the work of maintaining the populations. At each generation we haphazardly selected two males and two females from each successful family (those that had more than one larva survive to dispersal). These were haphazardly paired together, with the Control populations consistently comprising 35 pairs of beetles, and the No Care populations comprising 50 pairs. We used a larger number of pairs in the No Care treatment to accommodate the greater rate of brood failure associated with removing parents[47]. We make the reasonable assumption that the Control and No Care manipulations each increased predictably in the supply of care (see 'Results and discussion' for further details of parental assistance supplied before hatching). We could not force parents to supply care. However, the majority of broods in the Control treatment experienced at least 24 h of maternal care after hatching (Fig. 1). This is the period in which parental care has the greatest effect on measures of larval fitness[36]. Variation in the level of parental assistance available to larvae in penetrating the carcass was therefore greatly reduced in both treatments, compared with the natural variation shown in Fig. 1.

**Incisions by parents in the carcass before hatching**. In the 13th generation of experimental evolution, individuals were randomly paired within their respective replicate populations (cousins and siblings were not paired together) and placed in a breeding box (details in ref. [47]). Pairs were then provided with a recently defrosted mouse carcass (10–12 g) to breed upon. After ~53 h, i.e., the time at which parents are removed in the No Care populations, the carcasses they had prepared were examined and the presence of a parentally-derived feeding hole was noted, using the same method described above. Using a binomial test, we analysed whether the likelihood of a carcass bearing an incision differed among wild, No Care and Control populations.

We found that the No Care populations had a much larger proportion of carcasses with a parentally-derived incision (N1 = 60%, N2 = 62%) than either of the other two populations (results given in Main Text). The proportion of Control-prepared carcasses with an incision (F1 = 27%, F2 = 32%) was very similar to that seen when wild-caught beetles prepared carcasses (26%, Supplementary Fig. 2).

**The effect of parental incisions in the carcass on larval fitness**. This experiment focused on larvae from the Control and No Care populations after 13 generations of experimental evolution. A laboratory stock population prepared the

carcasses used in this experiment. The stock population was derived from a mixture of beetles caught in Gamlingay Woods, Waresley Woods and Byron's Pool, Cambridgeshire, UK. It was maintained using the same protocol as for the Control populations. Wild-caught beetles were interbred with the stock populations at every generation in the summer months, to maintain genetic diversity.

Carcasses were prepared by single virgin stock beetles, kept alone ($n = 56$ males, 57 females, 113 carcasses). Individuals were given a recently defrosted mouse carcass (8–14 g) and left to prepare the mouse for 68 h. We allowed a period longer than 53 h because single beetles take longer than pairs to complete carcass preparation[48]. Nevertheless, carcasses were removed from the parent before they could make an incision. The few carcasses that bore an incision were discarded. We divided the prepared carcasses between two treatments: Cut or Uncut. In the Cut treatment, we made an 8 mm incision into the thigh of the hind leg of the mouse. This part of the carcass was consistently exposed, which meant we did not need to unravel the now balled-up flesh to make the cut. The incision was as similar as possible to the cut made by the parental beetles. The Uncut carcasses were handled but left intact. This part of the methodology is very similar to the one performed in ref. [30].

Meanwhile, we paired beetles within each replicate of the Control (F1, $n = 15$; F2, $n = 35$) and No Care (N1, $n = 20$; N2, $n = 35$) treatments, using the procedure described above. Each pair was given a recently defrosted 24–26 g mouse to induce the laying of larger clutches. After 53 h, the carcass was removed and replaced with a small quantity of beef mince to ensure any newly-hatched larvae did not die from starvation. Breeding boxes were checked every eight hours for larvae. At hatching, we harvested newly-hatched, first-instar larvae for use in the experiment. First-instar larvae were collected into petri dishes each time we checked the boxes: one dish was for Control larvae, one was for No Care larvae. From each dish, ten larvae were chosen haphazardly from within each experimental population and placed directly on to a Cut or Uncut carcass. The experiment therefore had four treatments: Control larvae on a Cut carcass ($n = 24$); Control larvae on an Uncut carcass ($n = 23$); No Care larvae on a Cut carcass ($n = 33$); and No Care larvae on an Uncut carcass ($n = 33$). The larvae placed on each carcass received no post-hatching parental care. They were weighed and counted at dispersal, defined as occurring when two or more larvae were observed crawling away from the carcass[49].

The data were analysed with a generalised linear mixed-effects model, using the lme4 package[50] in R 3.3.0[51]. The number of surviving larvae was analysed using a generalised linear mixed-effects model with binomial error structure, since the data were bounded (by 0 and 10). We tested for the interaction between experimental care regime (i.e., the Control environment or No Care environment) and the carcass treatment (Cut or Uncut). Carcass mass and sex of the preparing beetle were included as covariates, with block as a random term. The number of successful broods (with at least one surviving larvae) was compared across treatments using a Fisher's Exact test.

We found that more larvae survived on a Cut carcass ($z = 9.074$, $P < 0.001$) than on an Uncut carcass, irrespective of whether they had been evolving under the Control or No Care treatment. Furthermore, the presence of a Cut increased larval survival to a similar degree, regardless of whether larvae originated from the Control or No Care populations (interaction between carcass treatment and experimental care regime: $z = 1.096$, $P = 0.273$). However, more No Care than Control larvae survived, whether the carcass was Cut or Uncut ($z = 8.032$, $P < 0.001$; Supplementary Fig. 3). Measuring success at the brood level, we found that no broods failed in the Cut treatment, regardless of whether larvae originated from the Control or No Care populations. However, in the Uncut treatment, No Care broods were more likely to survive than Control broods ($P < 0.001$). We draw two conclusions from this experiment. (1) Inserting an incision in the carcass prior to larval hatching promotes larval survival in general (see also ref. [30]). (2) No Care larvae are better adapted to life in a No Care environment than are Control larvae.

**Larval mandibles and body size in experimental populations**. In the 25th generation of experimental evolution, an experimental sub-population was put through a common garden environment in which all larvae received parental care, to reduce transgenerational effects and thereby expose any trait changes induced by the previous 24 generations of experimental evolution[52]. Using exactly the same procedures as described above for the wild populations, we dissected out mandibles from larvae in both replicate populations of the Control treatment (F1, $n = 45$; F2, $n = 37$) and both replicate populations of the No Care treatment (N1, $n = 46$; N2, $n = 40$). They were then measured in exactly the same way as wild larval mandibles. Two larvae were chosen haphazardly from each brood for measurement. We compared the scaling relationship (static allometry) between larval mandible length and body mass for third instar larvae from the Control and No Care populations, using the methodology outlined above.

We found that $\beta$ differed significantly between the Control and No Care treatments: there was a significant interaction between the experimental care regime and larval mass on the length of the larval mandibles (OLS: $t_{163} = -2.87$, $P = 0.005$; MA: $LR_1 = 9.65$, $P = 0.002$, Fig. 3). The static allometries in the Control and No Care treatments also differed significantly from a slope of zero (Supplementary Table 2): Control populations (OLS: $t_{80} = 11.39$, $P < 0.001$; MA: $r_{80} = 0.79$, $P < 0.001$); No Care populations (OLS regression: $t_{83} = 11.53$, $P < 0.001$;

MA regression: $r_{83} = 0.78$, $P < 0.001$). Both replicates, for both the Control (OLS: $t_{78} = -1.19$, $P = 0.24$) and No Care treatments (OLS: $t_{81} = -0.11$, $P = 0.91$), shared the same allometric slope.

**The adaptive value of mandible size in experimental larvae**. The protocol was identical to the experiment investigating the adaptive value of incisions in the carcass, described above, except that this time all the carcasses were Uncut (Control = 20, No Care = 35). Of the 55 pairs we set up, 39 yielded larvae that survived to dispersal. Each surviving larva was weighed at this point (Control = 12, No Care = 27). The mass of the smallest surviving larva from each brood derived from the Control experimental populations was compared with the mass of the smallest surviving larva from each brood derived from the No Care experimental populations, using a Kruskal–Wallis test in R. Since only ten larvae were placed onto each carcass, the total number of surviving larvae per brood was compared across treatments with a binomial test. The results are shown in Supplementary Fig. 3.

## Data availability

The experimental data that support the findings of this study have been deposited in figshare with the identifier https://doi.org/10.6084/m9.figshare.6355112[53].

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

## Acknowledgements

This project was funded by a European Research Council grant (310785 BALDWI-NIAN_BEETLES) and a Royal Society Wolfson Merit Award, both to R.M.K. We thank David Labonte and Ian Warren for comments on an earlier draft. We are very grateful to Sue Aspinall and Chris Swannack for maintaining the beetles in the laboratory.

## Author contributions

B.J.M.J., M.S. and R.M.K conceived the study. B.J.M.J., E.E., H.B.H., M.L., D.R. and A.D. collected data. B.J.M.J. analysed the data. B.J.M.J. and R.M.K. wrote the paper with input from all authors. All authors approved the final manuscript.

## Additional information

**Competing interests:** The authors declare no competing interests.

