## [Peer Review File · Nature Communications]

Reviewers' comments:

Reviewer #1 (Remarks to the Author):

In their manuscript "Variation in the supply of parental care drives rapid adaptive evolution of offspring morphology", Jarrett et al. use the burying beetle *Nicrophorus vespilloides* to investigate how selection stemming from social environments affects morphology. Specifically, they predict that larval mandibles, which are feeding structures, should evolve to be larger in beetles adapted to developing in the absence of parental care. Using experimental evolution lines, they find strong evidence for this prediction, forming the core result of the paper. The authors also present data showing that offspring performance improves when their nutritional resource (carcass) is cut, a clever experiment supporting the idea that a cutting instrument like a mandible may be relevant to fitness. The ideas and hypotheses underlying this manuscript are extremely interesting and valuable, and it has the potential to be a significant contribution to the field of social evolution. However, I have several concerns about the methodologies used and interpretation of data that I would like to see shored up.

Major comments:

1. In my opinion, the authors would have provided a substantially better test of their hypothesis had they measured mandible length in first or second instar larvae instead of only third instars. The presence or absence of parenting just might not be very important to third instars – Bartlett (1988 *Behav. Ecol. Sociobiol.*) reported that *N. vespilloides* barely feed third instars, and other work (e.g. Eggert et al. 1998 *Anim. Behav.*; Smiseth et al. 2003 *Proc. B*) suggests that removing parents has smaller effects on offspring performance at later developmental stages. Furthermore, because nobody can (I think) force an insect to provide care, parents in the "Full Care" lines used here probably in some cases deserted broods before the third instar. Both of these factors potentially undermine the authors' argument that predictability of parental care drives mandible evolution at this developmental stage. Though I certainly don't expect the authors to collect more data, I would like to see an explicit justification of why measuring third instar mandibles is appropriate here, as well as some discussion of how selection on morphology might vary at previous stages of development. Do the authors think selection is acting directly on third-instar mandibles, or might their results be a consequence of indirect selection targeting first or second-instar morphology?

2. I am not convinced that the data on wild-caught beetles shows that they adopt a bet-hedging strategy. If larvae produce large mandibles just in case the "worst-case scenario" happens and parents abandon, why would they subsequently evolve smaller mandibles in the No Care lines where they are adapted to this same worst-case scenario? The absence of any scaling relationship in Figure 2 is a surprising result, and I can sympathize with some difficulties in interpreting it. But I'm just not seeing how it argues for a bet-hedging strategy.

Following this, I don't believe that the conclusions presented in lines 252-260 are

appropriate. The authors would be better served by focusing discussion around their strongest result, the difference between their selection lines, rather than the more challenging interpretation of the data on wild-caught beetles.

3. In the paragraph from lines 206-231 (Figure 4), the authors claim to demonstrate explicitly that having larger mandibles improves the survival of smaller larvae in the absence parental care. But this is not what they show; rather, the data simply says that smaller larvae from No Care lines have higher survival than similarly sized larvae from Full Care lines. Mandible size is certainly one phenotypic difference that could be underlying this performance difference, but many other relevant larval traits might also have diverged between these lines (e.g. musculature, behavior, other mouthparts, etc.). To demonstrate a specific role of mandible size, the authors would need to perform a multiple regression disentangling the effects of body size and mandible size on survival.

Minor comments:

Lines 46-48: This idea may need fleshing out; I don't see how the two parts of this sentence relate.

Lines 108-110, 139-140: The term "Full Care" and its corresponding description are misleading, as it implies that all parents in this environment provide a maximum amount of care. The authors cannot know this, and just because the parents have not been physically removed by the experimenter doesn't mean they are actually providing care. This nuance should be made clear here, and correspondingly I prefer calling these "Control" lines, as the authors have done previously (Schrader et al. 2017 Proc. B).

Lines 177-179: A picture of a mandible, at least in Supplementary Material would be useful here for a few reasons. First, showing what the mandible looks like would give readers a clear picture of why someone might expect them to be involved in cutting into and feeding on a carcass. Second, for the methods, it would be helpful to see what landmarks the authors used for measurement. Currently, details on the measurement procedure are lacking and could not be reproduced (lines 303-308).

Line 256: I'd avoid the phrase "local adaptation", which conjures up nonexistent geographical differences.

Lines 334-335: Why exactly was mandible length cubed? What does it mean for body mass to be a cubic unit? The authors should provide justification and citation for this methodology if they continue to use it – I have not seen linear measurements cubed in similar allometric analyses using body mass. Does this qualitatively change the results?

Lines 417-430, 445-452, 487-507: Why are there statistical results presented in the Materials and Methods section?

Reviewer #2 (Remarks to the Author):

The authors use laboratory experimental evolution studies to investigate the naturally occurring phenotype of larger larval mandibles. They convincingly link the presence of the phenotype to the absence or reduction in parental care, wherein, larvae with larger mandibles are able to thrive in the absence of parental care. In contrast, in treatments where parental care was reliably present during early development, larval mandible size became smaller over time.

A related phenomenon, also in beetles, is egg cannibalism at hatching by group-living larvae in some chrysomelids. The puzzle in that group is, if it is better to live in a large group, why eat group members upon hatching? Combining field and laboratory studies, Breden and Wade (1989, *American Naturalist*) showed that small numbers of cannibalizing larvae were able to establish feeding sites on leaves of their host plants where similar numbers of non-cannibalistic larvae could not. Citing this work would help to put the author's findings in a larger, taxonomic and conceptual (cooperation v conflict) context.

The role of parents in worker size in the stem ant was also explored by Linksvayer (2006, 2007, 2008) using both intra-specific and inter-specific cross-fostering of brood and workers.

Reviewer #3 (Remarks to the Author):

The manuscript has an elegant set of experiments used to address the effects of parental care on offspring morphology, here the mouthpart size in burying beetles. The studies used include 1) quantification of the variation in parental care from wild-caught individuals in a laboratory setting, 2) a test of whether offspring can use cues of parental care to express adaptive phenotypic plasticity in mouthpart size, and 3) experimental evolution with two replicates of two care regimes – no care and full care, 4) an experiment testing the value of a cut made in the carrion by parents for offspring, and much more, including a series of measurements and comparisons of experimental individuals with each other and wild individuals.

This is a large body of work meticulously described and well written. It is inherently fascinating to see the ways in which parental care and offspring behavior evolve under experimental evolution conditions. Further, it provides an important advancement in our knowledge of the evolution in cooperative situations.

I have several comments intended to help make this manuscript as strong as it can possibly be. I want to emphasize that the number of comments simply reflects my enthusiasm for this nice piece of work.

Major comments:

1) The novelty of this work is sometimes overstated or misleading. For example: Lines 21-22 should be revised to include "experimentally," which would not negate the

existence and importance of other studies mentioned in Line 58.

Lines 206-208 is too strongly worded.

Lines 245-247. The way this is written, it sounds like a statement proven in multiple studies or across taxa. More caution is needed

2) Lines 84-94 describes an experiment using wild-caught individuals, brought to the lab, to test variation in the length of time parents spend with their offspring as well as the variation in the timing of hole-cutting. For the outside observer, it is not clear that this is a strong way to measure the length of time parents spend with offspring. Data should be shown that parents don't wander off occasionally, then come back. Here, they cannot come back, which seems very artificial. If these data do not yet exist, then their needs to be some discussion of the value and limitations of this metric. This issue ties back to the comment that the value of the work is sometimes overstated or misleading. Limitations needed to be stated more clearly throughout.

3) Lines 133, 212, 218, etc. throughout the smaller offspring receive disproportionate attention, yet the reason for this disproportionate attention is not clear until much later in the document (Lines 232-251). Even so, in Lines 232-251, the justification is unnecessarily dense, and it is unclear without checking the references whether the authors are making a general statement relevant across taxa or just relevant to these beetles. Further, readers that do not work on parental care may not automatically know that smaller individuals receive less care from parents. Much of this manuscript is predicated on this knowledge. Mention needs to be made earlier in the manuscript why the smaller individuals will receive so much of the focus. I encourage the authors to do so in a simple and straightforward fashion. It's also worth giving a little attention to the larger ones, including summarizing what all these experiments say about them.

4) Costs and benefits of mandible size are muddled. Lines 458-460. The idea that the small and large mouthparts are specialist strategies isn't really accurate. Presumably the large mouthparts are beneficial regardless, but they come with a developmental or maintenance cost. In fact, the authors mention in lines 470-473 that they expect Full Care should lead to relaxed selection for the large mouthparts, which should lead to greater variation in expression, presumably because greater size has a cost. When care is provided, the benefit of the larger mandibles is lost, leaving only the costs. The manuscript would benefit by clarifying the expected costs and benefits of large mandible size.

Writing concerns:

5) There are some small, but important writing issues. Line 34 "this" is vague and should be clarified. Line 36 "to be" is an awkward way to end this sentence.

6) The first sentence in Line 42 should be cut. It does little to enhance this work. It is needlessly vague and even distracting from the value of this study.

7) Throughout: "incision", "cut", and "hole" are used interchangeably. I recommend picking one word to describe it and sticking to it. Hole seemed particularly strange when used in the figure description.

8) Line 704. Not exactly "natural" variation, since it was measured in a highly-artificial laboratory environment.

Minor comments:

9) Lines 170-176. The authors predict that when selection is relaxed that mandibles should get smaller. It seems odd not to mention at this point that they should also become more variable in size.

10) Line 192. Do you think this is because of the incision?

11) Lines 299, 313, etc. Why are larvae sometimes measured at different ages? Could this affect conclusions drawn from the work?

12) Lines 440 – 441, etc. Why were two larvae chosen per brood?

Reviewers' comments:

Reviewer #1 (Remarks to the Author):

In their manuscript “Variation in the supply of parental care drives rapid adaptive evolution of offspring morphology”, Jarrett et al. use the burying beetle *Nicrophorus vespilloides* to investigate how selection stemming from social environments affects morphology. Specifically, they predict that larval mandibles, which are feeding structures, should evolve to be larger in beetles adapted to developing in the absence of parental care. Using experimental evolution lines, they find strong evidence for this prediction, forming the core result of the paper. The authors also present data showing that offspring performance improves when their nutritional resource (carcass) is cut, a clever experiment supporting the idea that a cutting instrument like a mandible may be relevant to fitness. The ideas and hypotheses underlying this manuscript are extremely interesting and valuable, and it has the potential to be a significant contribution to the field of social evolution. However, I have several concerns about the methodologies used and interpretation of data that I would like to see shored up.

Major comments:

1. In my opinion, the authors would have provided a substantially better test of their hypothesis had they measured mandible length in first or second instar larvae instead of only third instars. The presence or absence of parenting just might not be very important to third instars – Bartlett (1988 *Behav. Ecol. Sociobiol.*) reported that *N. vespilloides* barely feed third instars, and other work (e.g. Eggert et al. 1998 *Anim. Behav.*; Smiseth et al. 2003 *Proc. B*) suggests that removing parents has smaller effects on offspring performance at later developmental stages. Furthermore, because nobody can (I think) force an insect to provide care, parents in the “Full Care” lines used here probably in some cases deserted broods before the third instar. Both of these factors potentially undermine the authors’ argument that predictability of parental care drives mandible evolution at this developmental stage. Though I certainly don’t expect the authors to collect more data, I would like to see an explicit justification of why measuring third instar mandibles is appropriate here, as well as some discussion of how selection on morphology might vary at previous stages of development. Do the authors think selection is acting directly on third-instar mandibles, or might their results be a consequence of indirect selection targeting first or second-instar morphology?

The reviewer is correct in that selection is most likely acting on the first instar larvae that reach the carcass. However, we think our approach is justified for two key reasons: 1) previous work shows a larva with relatively large mandibles in its first instar will also have relatively large mandibles in its third instar (Ruzicka 1992, Benowitz et al. *bioRxiv*). 2) developmental and genetic studies of other insects indicate that the mechanisms governing mouthpart size do not differ between larval instars (Alvarez et al 2017). Thus, we should be able to observe the correlated evolution of the mandibles of the third instar when selection has acted on the same developmental mechanisms that dictate mandible size in the first instar (L126-134). We chose not to measure the mandibles of first instar larvae because we found it difficult to obtain precise measurements.

Turning to the point about parental care, again we agree. An important point, which we now emphasise to a greater degree, is that our manipulation predictably reduced the extent of carcass preparation by parents: Full Care/ Control parents were all able to insert an incision in the carcass; most No Care parents – at least initially- were not (see Figure 1). Most Full Care / Control broods experienced at least 24h of maternal care (Figure 1). None of the No Care brood experienced any post-hatching care. Therefore even though we could not induce parents to supply care, we did predictably manipulate the extent to which parents helped their offspring, and we reduced variation in the extent of care, relative to the natural population, in both treatments (L154-159, L184-186, L376-384, and also Figure 1 and legend, L639-646).

2. I am not convinced that the data on wild-caught beetles shows that they adopt a bet-

hedging strategy. If larvae produce large mandibles just in case the “worst-case scenario” happens and parents abandon, why would they subsequently evolve smaller mandibles in the No Care lines where they are adapted to this same worst-case scenario? The absence of any scaling relationship in Figure 2 is a surprising result, and I can sympathize with some difficulties in interpreting it. But I’m just not seeing how it argues for a bet-hedging strategy.

We have modified our interpretation of the wild-caught data now. We suggest that the variation in the larval mandible scaling relationship is due to variable levels of parental care, and discuss possible mechanisms by which this association might arise (L245-260).

Following this, I don’t believe that the conclusions presented in lines 252-260 are appropriate. The authors would be better served by focusing discussion around their strongest result, the difference between their selection lines, rather than the more challenging interpretation of the data on wild-caught beetles.

We have limited our discussion of bet-hedging now to the concluding discussion (L245-249, L254-L255)

3. In the paragraph from lines 206-231 (Figure 4), the authors claim to demonstrate explicitly that having larger mandibles improves the survival of smaller larvae in the absence parental care. But this is not what they show; rather, the data simply says that smaller larvae from No Care lines have higher survival than similarly sized larvae from Full Care lines. Mandible size is certainly one phenotypic difference that could be underlying this performance difference, but many other relevant larval traits might also have diverged between these lines (e.g. musculature, behavior, other mouthparts, etc.). To demonstrate a specific role of mandible size, the authors would need to perform a multiple regression disentangling the effects of body size and mandible size on survival.

This is an excellent point and we have changed the description of the experiment and our interpretation of the results accordingly. Specifically, we now acknowledge that we cannot attribute survival differences among larvae to the mandibles alone. Indeed, we know from our own unpublished work that other larval traits have evolved in the No Care populations and that they are adaptive in promoting larval survival - and we have now added this information to the text (L236-244)

Minor comments:

Lines 46-48: This idea may need fleshing out; I don’t see how the two parts of this sentence relate.

We have completely re-written this section

Lines 108-110, 139-140: The term “Full Care” and its corresponding description are misleading, as it implies that all parents in this environment provide a maximum amount of care. The authors cannot know this, and just because the parents have not been physically removed by the experimenter doesn’t mean they are actually providing care. This nuance should be made clear here, and correspondingly I prefer calling these “Control” lines, as the authors have done previously (Schrader et al. 2017 Proc. B).

We agree that the term ‘Full Care’ is insufficiently nuanced. Therefore we have changed our description of this treatment to ‘Control’, as the reviewer suggests. We also explain in detail exactly what type and level of care broods in each of these treatments might expect (L154-159, L184-186, L376-384) and why they are still more predictable (in their different ways) than might be seen in wild populations (L187-190)

Lines 177-179: A picture of a mandible, at least in Supplementary Material would be useful here for a few reasons. First, showing what the mandible looks like would give readers a clear picture of why someone might expect them to be involved in cutting into and feeding on a carcass. Second, for the methods, it would be helpful to see what landmarks the authors used for measurement. Currently, details on the measurement procedure are lacking and could not be reproduced (lines 303-308).

We have added in an image of a mandible as suggested (Supplementary Figure One, L696-700).

Line 256: I'd avoid the phrase "local adaptation", which conjures up nonexistent geographical differences.

We have removed this phrase.

Lines 334-335: Why exactly was mandible length cubed? What does it mean for body mass to be a cubic unit? The authors should provide justification and citation for this methodology if they continue to use it – I have not seen linear measurements cubed in similar allometric analyses using body mass. Does this qualitatively change the results?

We have cubed the mandible length to maintain dimensionality with body mass. This does not alter the results; it merely simplifies interpretation of the allometric slope. After a transformation of this sort, a slope of 1 indicates isometry. We now explain this in the Materials and Methods (L354-357).

Lines 417-430, 445-452, 487-507: Why are there statistical results presented in the Materials and Methods section?

This arises as a result of the journal's preferred format of integrating the Results and the Discussion. Some results are important to report to the specialist reader but distract from the thread of the argument we wish to run through the Discussion for the more general reader. These are the results we have reported in the Materials and Methods section. Although we have not revised the manuscript in response to this comment, we are happy to do so if the editor thinks it is necessary.

Reviewer #2 (Remarks to the Author):

The authors use laboratory experimental evolution studies to investigate the naturally occurring phenotype of larger larval mandibles. They convincingly link the presence of the phenotype to the absence or reduction in parental care, wherein, larvae with larger mandibles are able to thrive in the absence of parental care. In contrast, in treatments where parental care was reliably present during early development, larval mandible size became smaller over time.

A related phenomenon, also in beetles, is egg cannibalism at hatching by group-living larvae in some chrysomelids. The puzzle in that group is, if it is better to live in a large group, why eat group members upon hatching? Combining field and laboratory studies, Breden and Wade (1989, American Naturalist) showed that small numbers of cannibalizing larvae were able to establish feeding sites on leaves of their host plants where similar numbers of non-cannibalistic larvae could not. Citing this work would help to put the author's findings in a larger, taxonomic and conceptual (cooperation v conflict) context.

We agree that this paper is highly relevant to our work. In fact, it is particularly relevant to a parallel study (described in a different paper that is currently in revision at Proc R Soc B), which focuses on the experimental evolution of different larval adaptations. In this parallel study, the conflict/ cooperation tension within the brood is much more pertinent to the discussion of our results - and we will certainly reference Breden and Wade 1989 there.

The role of parents in worker size in the stem ant was also explored by Linksvayer (2006, 2007, 2008) using both intra-specific and inter-specific cross-fostering of brood and workers.

Yes – this is a good point. We now have broadened our introduction to include references to this work (L49-51)

Reviewer #3 (Remarks to the Author):

The manuscript has an elegant set of experiments used to address the effects of parental care on offspring morphology, here the mouthpart size in burying beetles. The studies used include 1) quantification of the variation in parental care from wild-caught individuals in a laboratory setting, 2) a test of whether offspring can use cues of parental care to express adaptive phenotypic plasticity in mouthpart size, and 3) experimental evolution with two replicates of two care regimes – no care and full care, 4) an experiment testing the value of a cut made in the carrion by parents for offspring, and much more, including a series of measurements and comparisons of experimental individuals with each other and wild individuals.

This is a large body of work meticulously described and well written. It is inherently fascinating to see the ways in which parental care and offspring behavior evolve under experimental evolution conditions. Further, it provides an important advancement in our knowledge of the evolution in cooperative situations.

I have several comments intended to help make this manuscript as strong as it can possibly be. I want to emphasize that the number of comments simply reflects my enthusiasm for this nice piece of work.

Major comments:

1) The novelty of this work is sometimes overstated or misleading. For example: Lines 21-22 should be revised to include “experimentally,” which would not negate the existence and importance of other studies mentioned in Line 58.

We have re-written the abstract

Lines 206-208 is too strongly worded.

We have re-written this section

Lines 245-247. The way this is written, it sounds like a statement proven in multiple studies or across taxa. More caution is needed

We have removed these lines

2) Lines 84-94 describes an experiment using wild-caught individuals, brought to the lab, to test variation in the length of time parents spend with their offspring as well as the variation in the timing of hole-cutting. For the outside observer, it is not clear that this is a strong way to measure the length of time parents spend with offspring. Data should be shown that parents don't wander off occasionally, then come back. Here, they cannot come back, which seems very artificial. If these data do not yet exist, then their needs to be some discussion of the value and limitations of this metric. This issue ties back to the comment that the value of the work is sometimes overstated or misleading. Limitations needed to be stated more clearly throughout.

This is very helpful. We can now see why readers who are not familiar with burying beetle natural history might think there are problems with the design of this experiment. We now provide greater detail to explain why our experimental design accurately assesses variation in the duration of parental care under standardised conditions in the lab (L72-86). In brief, observations on wild beetles in the field show that a) they abandon larvae before larval development is complete and b) once desertion has occurred, parents do not return (Scott

and Traniello 1990). Similar patterns have also been previously observed in the lab – even when beetles do have the opportunity to return they do not (Scott 1998).

3) Lines 133, 212, 218, etc. throughout the smaller offspring receive disproportionate attention, yet the reason for this disproportionate attention is not clear until much later in the document (Lines 232-251). Even so, in Lines 232-251, the justification is unnecessarily dense, and it is unclear without checking the references whether the authors are making a general statement relevant across taxa or just relevant to these beetles. Further, readers that do not work on parental care may not automatically know that smaller individuals receive less care from parents. Much of this manuscript is predicated on this knowledge. Mention needs to be made earlier in the manuscript why the smaller individuals will receive so much of the focus. I encourage the authors to do so in a simple and straightforward fashion. It's also worth giving a little attention to the larger ones, including summarizing what all these experiments say about them.

We have removed all reference to smaller larvae until the run-up to the final experiment. Note that we focus on smaller larvae there not because they may (or may not) receive preferential feeding from parents but because this size class of larvae differs most in the relative size of their mandibles across populations (Figure 3).

4) Costs and benefits of mandible size are muddled. Lines 458-460. The idea that the small and large mouthparts are specialist strategies isn't really accurate. Presumably the large mouthparts are beneficial regardless, but they come with a developmental or maintenance cost. In fact, the authors mention in lines 470-473 that they expect Full Care should lead to relaxed selection for the large mouthparts, which should lead to greater variation in expression, presumably because greater size has a cost. When care is provided, the benefit of the larger mandibles is lost, leaving only the costs. The manuscript would benefit by clarifying the expected costs and benefits of large mandible size.

We have clarified this point in the final paragraph (L261-276).

Writing concerns:

5) There are some small, but important writing issues. Line 34 "this" is vague and should be clarified. Line 36 "to be" is an awkward way to end this sentence.

We have re-written this section

6) The first sentence in Line 42 should be cut. It does little to enhance this work. It is needlessly vague and even distracting from the value of this study.

We have removed the sentence as suggested.

7) Throughout: "incision", "cut", and "hole" are used interchangeably. I recommend picking one word to describe it and sticking to it. Hole seemed particularly strange when used in the figure description.

Good point. For consistency, we have revised the paper so that "feeding incision" is used when referring to holes made in the carcass by beetles and "cut" is used when referring to our experiment where we manipulated the cut – it was not parentally-derived.

8) Line 704. Not exactly "natural" variation, since it was measured in a highly-artificial laboratory environment.

Good point. We have clarified the meaning here by referring to it as the behaviour shown by wild-caught individuals when bred under standardised laboratory conditions (L638-641).

Minor comments:

9) Lines 170-176. The authors predict that when selection is relaxed that mandibles

should get smaller. It seems odd not to mention at this point that they should also become more variable in size.

We have clarified our predictions here (L196-199).

10) Line 192. Do you think this is because of the incision?

We think it is partially to do with the timing of the feeding incision by parents. The wild population creates a feeding incision prior to larval hatching at the same frequency as the Control population. There may also be other adaptations that have occurred in the No Care parents during carcass preparation that have also led to the erosion of the No Care mandible allometry, but we think the incision plays a major role.

11) Lines 299, 313, etc. Why are larvae sometimes measured at different ages? Could this affect conclusions drawn from the work?

The larvae were in fact all measured at the same age. The third instar larvae disperse from the carcass eight days after pairing. We have clarified this point (L317-319).

12) Lines 440 – 441, etc. Why were two larvae chosen per brood?

To ensure we had replicate measurements from each brood. We now explain this (L319-320).

Reviewers' comments:

Reviewer #1 (Remarks to the Author):

In their revision, Jarrett et al. have done an excellent job addressing all of the concerns and questions I had regarding the original manuscript. In my opinion, it is a substantially improved work, and I have no further points that I feel need to be addressed.

Reviewer #3 (Remarks to the Author):

The authors have done a good job addressing reviewer comments, and the manuscript is much improved. In addition to the major changes already completed, I have noticed a few areas for improvement:

- 1) The picture of the mandible would be improved by including along side it a picture of a larva of this species, showing the location of the mandible and its size in relation to the whole animal.
- 2) The renaming of the "Full Care" lines to "Control" lines is a big improvement, given that the actual amount of care given within these lines is both variable and unknown. This point is important to be clear about, and I suggest adding a statement after the sentence finishing in Line 117 that the actual level of care by parents is unknown and is likely variable (though more than the "No Care" lines)
- 3) Line 215. Explanation is needed on why wild-type larvae should have even larger mandibles for a given body size. Could this be a spurious result? Has it been tested statistically?
- 4) Line 219. The statement "we sought evidence" is extremely problematic. Presumably the authors were not just searching for evidence to support a pet hypothesis, but this is what it sounds like. It's crucial to show an unbiased approach to collecting data. Please reword this statement.
- 5) Line 242. "Other larval adaptations" is very tantalizing, and a couple of examples would be most kind to provide the reader if possible.
- 6) Line 249. Is there heritability in parental care?
- 7) Lines 251-253. I recommend not starting sentences with long clauses. It's more difficult to read and follow.
- 8) Line 259. "in part" should be added after "explains"

Reviewer 3's comments

The authors have done a good job addressing reviewer comments, and the manuscript is much improved. In addition to the major changes already completed, I have noticed a few areas for improvement:

1) The picture of the mandible would be improved by including along side it a picture of a larva of this species, showing the location of the mandible and its size in relation to the whole animal.

Good suggestion. We have added in an SEM image of the head of a larva in Supplementary Figure 1.

2) The renaming of the “Full Care” lines to “Control” lines is a big improvement, given that the actual amount of care given within these lines is both variable and unknown. This point is important to be clear about, and I suggest adding a statement after the sentence finishing in Line 117 that the actual level of care by parents is unknown and is likely variable (though more than the “No Care” lines)

We have done this (Lines 123-126).

3) Line 215. Explanation is needed on why wild-type larvae should have even larger mandibles for a given body size. Could this be a spurious result? Has it been tested statistically?

We explain why we think this result arose (highly unpredictable care in the Wild populations v more predictable care in the experimental populations) (Lines 222-225); it is not a spurious result ie we have tested it statistically (Supplementary table 2)

4) Line 219. The statement “we sought evidence” is extremely problematic. Presumably the authors were not just searching for evidence to support a pet hypothesis, but this is what it sounds like. It's crucial to show an unbiased approach to collecting data. Please reword this statement.

We have reworded this sentence (Line 225).

5) Line 242. “Other larval adaptations” is very tantalizing, and a couple of examples would be most kind to provide the reader if possible.

We have added in some examples (Lines 249-250).

6) Line 249. Is there heritability in parental care?

We are not sure how this is relevant.

But anyway, the answer is to some extent, yes. Walling et al. 2008 PNAS showed that there are small to moderate heritabilities of both direct and indirect care. However, no work has been done on the heritability of the extent of care that parents provide.

7) Lines 251-253. I recommend not starting sentences with long clauses. It's more difficult to read and follow.

We are unsure about the long clause reviewer 3 is referring to. That line is:
“However, the mechanism that maintains high levels of variation in this scaling relationship within wild populations (Fig. 2) remains to be determined in future work.”

8) Line 259. “in part” should be added after “explains”

We have added this to the sentence (Line 287).